# Advances in the Development of SARS-CoV-2 Mpro Inhibitors

**DOI:** 10.3390/molecules27082523

**Published:** 2022-04-14

**Authors:** Laura Agost-Beltrán, Sergio de la Hoz-Rodríguez, Lledó Bou-Iserte, Santiago Rodríguez, Adrián Fernández-de-la-Pradilla, Florenci V. González

**Affiliations:** 1Departament de Química Inorgànica i Orgànica, Universitat Jaume I, 12080 Castelló, Spain; lagost@uji.es (L.A.-B.); sdelahoz@uji.es (S.d.l.H.-R.); lbou@uji.es (L.B.-I.); rodrigue@uji.es (S.R.); 2Departament de Química Física i Analítica, Universitat Jaume I, 12080 Castelló, Spain; delaprad@uji.es

**Keywords:** COVID-19, main protease, Mpro, inhibitors

## Abstract

Since the outbreak of COVID-19, one of the strategies used to search for new drugs has been to find inhibitors of the main protease (Mpro) of the virus SARS-CoV-2. Initially, previously reported inhibitors of related proteases such as the main proteases of SARS-CoV and MERS-CoV were tested. A huge effort was then carried out by the scientific community to design, synthesize and test new small molecules acting as inactivators of SARS-CoV-2 Mpro. From the chemical structure view, these compounds can be classified into two main groups: one corresponds to modified peptides displaying an adequate sequence for high affinity and a reactive warhead; and the second is a diverse group including chemical compounds that do not have a peptide framework. Although a drug including a SARS-CoV-2 main protease inhibitor has already been commercialized, denoting the importance of this field, more compounds have been demonstrated to be promising potent inhibitors as potential antiviral drugs.

## 1. Introduction

The outbreak of COVID-19 has paralyzed the globe. As of 22 March 2022, the total confirmed cases are nearly 471 million, and the total deaths more than 6 million worldwide [1]. The massive vaccination campaign for COVID-19 in many countries is expected to result in herd immunity. These vaccines target the spike protein of the SARS-CoV-2 virus [2]. However, the spike protein is highly mutable, as confirmed by new SARS-CoV-2 variants. The spike protein from SARS-CoV-2 shares 76% sequence identity with the spike protein from SARS-CoV [3]. Although booster vaccines might be developed for new variants, small molecule antivirals against less mutable targets will be more successful than a vaccine for the treatment of patients with severe symptoms and also for prevention. In general terms, administration, delivery, storage and production of small molecules are easier than vaccines. The search for new small molecules as drugs of COVID19 includes proteolytic targets in SARS-CoV-2 infection: two viral proteases called main protease (Mpro) and papain-like protease (PLpro); and three human proteases known as transmembrane protease serine 2 (TMPRSS2), cathepsin L and furin. Unlike spike, the Mpro enzyme has a highly conserved gene. The sequence identity between SARS-CoV and SARS-CoV-2 for Mpro is 96%, much higher than spike protein (76%) or the overall 82% genome sequence identity between both viruses.

During the replication cycle, the coronaviruses express two overlapping polyproteins (pp1a and pp1b) and four structural proteins from the viral RNA. The polyproteins pp1a and pp1b liberate the mature viral proteins required for replication after being processed by two cysteine proteases coded in the viral genome: the main protease (Mpro) is also known as 3-chymotrypsin-like protease (3CLpro) which carries out most of the cleavages; and papain-like protease (PLpro). The cleavage site of Mpro is between a glutamine at the P1 site and a small residue at P1′, such as alanine. The P2 site is commonly occupied by a leucine residue. 

This review deals with the progress made in the search for inhibitors of SARS-CoV-2 Mpro. We will focus on the SARS-CoV-2 Mpro inhibitors that have been chemically synthesized and biologically tested.

## 2. Peptidyl SARS-CoV-2 Mpro Inhibitors

In the search for SARS-CoV-2 Mpro inhibitors, repurposing of approved inhibitors of similar proteases was the first approach. In addition, the inhibitors of SARS-CoV Mpro and MERS-CoV Mpro—both enzymes structurally similar to SARS-CoV-2 Mpro—were tested.

In April 2020, a study to identify Mpro inhibitors by combining structure-based virtual and high-throughput screening was performed. More than 10,000 compounds were assayed, including approved drugs, drug candidates in clinical trials, and other bioactive compounds [4]. The Michael acceptor compound **N3**, previously reported as an inhibitor of Mpro enzymes of SARS-CoV and MERS-CoV, was identified as a time-dependent inhibitor of SARS-CoV-2 Mpro with k_obs_/[I] of 11,300 M^−1^ s^−1^ (Figure 1). The compound **N3** gave an antiviral effect in SARS-CoV-2-infected Vero cells with an EC_50_ value of 16.77 μM. 

The crystal structure of SARS-CoV-2 Mpro in complex with the inhibitor was resolved (2.1 Å resolution) (PDB 6LU7). The electron density showed the formation of a covalent bond between the Cβ atom of the vinyl group and the sulfur atom of Cys145 (1.8 Å C-S distance), confirming that compound **N3** acts as a Michael-type inhibitor. The lactam ring of the inhibitors accommodated into the S1 site formed by the side chains of residues 140, 142, 166, 163 and 172 of protomer A of the structure, which also includes two ordered water molecules. At the S2 site, the isobutyl group of Leu was surrounded by residues 41, 49, 54, 165 and 187. At the S3 site, the isopropyl group of Val was solvent-exposed, and the alanine accommodated at the S4 side surrounded by residues 165, 167, 185, 192 and 189 (Figure 1) [5].

Also in April 2020, R. Hilgenfeld et al. reported the ketoamides **13a** and **13b** (Figure 2) to be SARS-CoV-2 Mpro inhibitors [6]. The design of these compounds was based upon a previously reported compound active against MERS-CoV and SARS-CoV. The P2-P3 amide bond in the new compounds had a pyridone to prevent cleavage by proteases, hence improving the half-life of the compounds in plasma. Inhibitor **13b** was active against SARS-CoV-2 Mpro (IC_50_ = 0.67 μM), and inhibited RNA replication (EC_50_ = 1.75 μM) and infected Calu-3 cells infected with SARS-CoV-2 (EC_50_ = 4–5 μM).

The crystal structures of the complex formed between compound **13b** and SARS-CoV-2 Mpro show a thiohemiketal resulting from the nucleophilic attack of the thiol group of Cys145 to the keto carbonyl group of the ketoamide moiety of **13b** (PDB 6Y2F). Thiohemiketal was stabilized in the oxyanion hole by hydrogen bonding with amides of residue His41, and the amide oxygen acted as a hydrogen-acceptor group from amides of Gly143, Cys145 and Ser144. The S1 site accommodated the lactam ring by hydrogen-bonding with Phe140 and the Glu166 carboxylate, and the carbonyl oxygen accepted a hydrogen bond from the imidazole of His163. The P2 cyclopropyl methyl moiety of **13b** fit into the S2 subsite, shrunk as compared to the complex between **13a** and the SARS-CoV Mpro having a cyclohexyl methyl at the P2 site. The carbonyl oxygen of the pyridone-ring hydrogen-bonded with the amide of the main chain of Glu166 whilst the nitrogen of the pyridone ring did not participate in any hydrogen bond. The Boc group did not occupy the S4 site but was slightly displaced towards Pro168 (Figure 2) [5].

In June 2020, H. Liu, H. Yang, L. Zhang, Y. Xu et al. studied the activity of compounds **11a** and **11b** as inhibitors of the protease (Figure 3). Both compounds had an aldehyde group as a warhead to be attacked by the cysteine 145 of the active center and the glutamine surrogate at the P1 site. A cyclohexyl or 3-fluorophenyl ring was introduced into the P2 position to test the influence of the ring. For both inhibitors, an indole group was introduced in the P3 site to form new hydrogen bonds with S4 and improve drug-like properties. Compounds were active, displaying IC50 values of 0.053 μM and 0.040 μM for **11a** and **11b**, respectively [7].

The crystal structure of the complex between SARS-CoV-2 Mpro and inhibitor **11a** was determined at 1.5 Å resolution (PDB 6MOK). In the crystal, an asymmetric unit contains one molecule and the crystal belonged to the space group C2. Protomers A and B associated into a homodimer with a two-fold symmetry axis. The compound **11a** extended along the active center. The aldehyde carbon of **11a** and the sulfur of Cys145 form a C-S covalent bond (1.8 Å), and the aldehyde oxygen hydrogen-bonded with the amide of Cys145. At the S1 site, the oxygen of the lactam interacted with the imidazole ring of His163 whilst the lactam NH group was hydrogen-bonding with the main chain of Phe140. The cyclohexyl group at P2 interacted with Met49, Tyr54, Met165, Asp187, and Arg188 by hydrophobic interactions and stacked with the imidazole ring of His41 occupying the S2 site. The indole group of **11a** at P3 was solvent-exposed and formed a hydrogen bond with Glu166 and hydrophobic interactions with Pro168 and Gln189 (Figure 3). The crystal structure of **11b** in complex with SARS-CoV-2 was very similar, despite small differences in the binding mode.

Inhibitors **11a** and **11b** were assayed in cell culture infected with SARS-CoV-2 virus, giving EC_50_ values of 0.53 mM and 0.72 mM, respectively. Neither compound was toxic (CC_50_ > 100 mM). Both compounds showed good pharmacokinetic properties. An in vivo toxicity study over rats and dogs gave no obvious toxicity in either group, revealing **11a** as a good candidate for further clinical study.

In another study, a list of 55 known peptidyl compounds acting as inhibitors of proteases (proteasome, aspartyl, serine, cysteine and metalloproteases) were tested against SARS-CoV-2 Mpro using the FRET-based enzymatic assay [8].

Among all tested compounds, four of them (FDA-approved HCV drug boceprevir, GC-376 and calpain inhibitors II and XII in preclinical assays) (Figure 4)) had single-digit to submicromolar IC_50_ values against SARS-CoV-2 Mpro and inhibited SARS-CoV-2 viral replication in cell culture with low-micromolar EC_50_ values. They also were low-toxic with CC_50_ > 100 mM for boceprevir, GC-376 and calpain inhibitors II, and calpain inhibitor XII CC_50_ > 50 mM. The warhead of calpain inhibitor XII and boceprevir is a ketoamide and the warhead of calpain inhibitor II. Inhibitor GC-376 is a bisulfite adduct acting as a prodrug of the active aldehyde form. 

The crystal structure of SARS-CoV-2 Mpro with inhibitor GC-376 was solved (2.15 Å resolution) revealing a hemithioketal resulting from the combination between the aldehyde warhead and Cys145 (PDB 6WTT). GC-376 mimicked the substrate of Mpro by hydrogen bonding along the active site. At the S1 site, the glutamine surrogate γ-lactam ring formed hydrogen bonds with the His163 and Glu166 side chains and the main chain of Phe140. The isobutyl moiety of a leucine residue at P2 position occupied the hydrophobic site formed by His41, Met49, and Met169 (Figure 4) [5]. In other reported Mpro inhibitors, the P2 position was also an isobutyl (**N3**), cyclopropyl (**13b**), cyclohexyl (**11a**), and 3-fluorophenyl (**11b**). The carbamate group in GC-376 interacted with Glu166 through a hydrogen bond, and the benzyl group of the carbamate complemented along the aliphatic S4 site through hydrophobic interactions.

More recently, GC-376 has been demonstrated to be suitable for SARS-CoV-2 therapy through a mouse model [9].

J. Qiao et al. reported the design and synthesis of 32 new SARS-CoV-2 Mpro inhibitors [10]. The chemical structure of the new compounds had an aldehyde as a warhead, the already used glutamine surrogate at P1, a bicycloproline-containing P2 derived from either boceprevir or telaprevir, and the residue at P3, which was allowed to change (Figure 5). The IC_50_ values for in vitro activity ranged from 7.6 to 748.5 nM (24 compounds displayed two-digit nanomolar IC_50_ values, and three of them exhibited single-digit values). The crystal structure of the complex formed between Mpro and inhibitor MI-23 (IC_50_ = 7.6 nM) was determined (PDB 7D3I). As expected, the inhibitor fit into the active site and the carbon of the warhead aldehyde formed a covalent bond with the sulfur atom of residue Cys145. The oxygen of the aldehyde formed two hydrogen bonds with the main-chain amides of Cys145 and Gly143 in the “oxyanion hole” (Figure 5) [5]. The gamma-lactam ring at P1 formed two hydrogen bonds with His163 and Phe140 inserting deeply into the S1 pocket. The rigid P2 bicycloproline adopted the trans-exo conformation, causing the bicycloproline group to point towards the S2 pocket in order to interact hydrophobically with residues Met165, Gln189, His41, Met49, Asp187 and Arg188. The 1-ethyl-3,5-difluorobenzene moiety at P3 extended along the S4 site through hydrophobic interactions with Gln189, Leu167 and Pro168. Compounds were active in cells with EC_50_ values ranging from 0.53 to 30.49 mM and showed no toxicity on cells (CC_50_ > 500 mM). Two inhibitors were assayed on rats. Two compounds showed relatively good pharmacokinetics with oral bioavailability above 10%.

In September 2020, scientists from the Pfizer company reported compound PF-00835231 (Figure 6) as a potent irreversible inhibitor of SARS-CoV-2 Mpro [11]. The compound had been previously identified for the treatment of SARS-CoV, and given that SARS-CoV Mpro and SARS-CoV-2 Mpro share 96% identity overall and 100% identity in the active site, it was repurposed. Compound PF-00835231 is a hydroxy ketone inhibitor. Then, the novel phosphate prodrug PF-07304814 was also described to increase the potential for the intravenous treatment of COVID-19 disease [12].

Compounds were tested in vitro in two cell lines (kidney and lung). For the kidney-cell assay, higher activity of **PF-00835231** was observed when a P-glycoprotein (P-gp) transporter inhibitor was used (EC_50_ up to 0.23 μM with a concentration of 2 μM of P-gp inhibitor). No toxicity was detected. The metabolic-stability studies of **PF-00835231** indicated that it provides a low risk of drug–drug interactions on coadministration with other drugs. Preclinical in vitro and in vivo assays showed conversion of phosphate **PF-07304814** into **PF-00835231**. Prodrug **PF-07304814** exhibited a good nonclinical safety profile. Phase 1 clinical studies are in progress.

With the objective of making an oral drug for COVID-19, scientists from the Pfizer company studied new Mpro inhibitors with better oral absorption than compound **PF-00835231**. The warhead of the previous **PF-00835231** would be changed since hydroxymethyl ketone moiety is a hydrogen-bond donor, and it is known that these types of groups correlate with poor bioavailability. Two already-known warheads of cysteine proteases were considered for the new Mpro inhibitors: benzothiazol-2-yl ketone and nitrile [13]. The first inhibitor of the series resulted from the substitution of the hydroxymethyl ketone warhead of **PF-00835231** by a nitrile group. The resulting compound (**A**, Figure 7) showed higher rat oral bioavailability but less potency against SARS-CoV-2 Mpro and lower antiviral activity (Figure 7). If a cyclic leucine mimetic (6,6-dimethyl-3-azabicyclo[3.1.0]hexane) was introduced at the P2 position, and the group at P1′ acting as a warhead was a benzothiazolyl ketone, then the permeability was high but the SARS-CoV-2 Mpro potency and the metabolic stability were lower (compound **B**, Figure 7). Introducing a methanesulfonamide group at P3 (compound **C**, Figure 7) instead of an indole ring increased hydrogen bonding with the Glu166, hence improving potency and antiviral activity (Figure 7). Even higher antiviral activity was obtained when a trifluoroacetamide group was introduced at P3 (compound **D**, Figure 7). Finally, the best candidate (**PF-07321332**) was found when a nitrile was introduced as a warhead over this scaffold, thus giving rise to compound **PF-07321332**. This compound is a potent inhibitor of SARS-CoV-2 Mpro with improved antiviral activity (Figure 7). It inhibits SARS-CoV-2 Mpro in a reversible mode, as demonstrated by competitive assays with an irreversible inhibitor. This compound was chosen as a clinical candidate based on a reduced tendency to epimerization at the P1 stereocenter, ease of synthetic scale-up and enhanced solubility. This compound, named nirmatrelvir, is now commercialized as a drug for COVID-19 disease combined with protease inhibitor ritonavir and sold under the brand name Paxlovid.

## 3. Non-Peptidyl SARS-CoV-2 Mpro Inhibitors

In a study to identify Mpro inhibitors, more than 10,000 compounds were assayed, including approved drugs, drug candidates in clinical trials and other bioactive compounds [4,14]. Among all tested compounds, 2-phenyl-1,2-benzoselenazol-3-one (known as ebselen) was identified as a potent inhibitor of SARS-CoV-2 Mpro (IC_50_ of 0.67 μM) (Figure 8). Ebselen is a low-toxicity compound which has been previously reported as having anti-inflammatory, antioxidant and cytoprotective properties. The compound showed the strongest antiviral effect at a concentration of 10 μM treatment in SARS-CoV-2-infected Vero cells (EC_50_ = 4.67 μM). Later, Weglarz-Tomczyk et al. identified ebselen as an inhibitor of SARS-CoV-2 PLpro protease (IC_50_ = 2.26 μM) [15].

The mode of action of ebselen was initially studied by molecular dynamics [16]. They found two highly probable interaction sites between SARS-CoV-2 Mpro and ebselen. One is in the active site and the other one is in the region between the II and III domains, which is essential for Mpro dimerization

Interestingly, the crystal structure of the complexes formed between ebselen and its analog **MR6-31-2** with SARS-CoV-2 Mpro were solved (PDB 7BAK and 7BAL, respectively). They showed the selenium atom to be bound to the sulfur atom of Cys145 but without the organic framework of ebselen (Figure 9) [5,17]. A mechanism was suggested through the attack of the sulfur to the selenium followed by hydrolysis, releasing a salycinilide by-product detected by mass spectrometry (Figure 9). This mechanism has since been studied through a combined Docking and Density Functional Theory (DFT) approach [18].

K-W. Yang et al. performed a FRET screening study of 36 compounds structurally derived from ebselen and ebsulfur, which afforded compounds **1i** and **2k** as the most active ones (Figure 10) [19]. Interestingly, both compounds displayed a furan ring as a substituent. However, these compounds were unable to affect inhibition in the presence of DTT. In addition, J. Wang et al. studied the mechanism of action of ebselen and another five previously reported Mpro inhibitors (disulfiram, carmofur, PX-12, tideglusib and shikonin) and concluded that the inhibition is abolished or greatly reduced with the addition of reducing reagent DTT, determining that they are nonspecific promiscuous SARS-CoV-2 main protease inhibitors [20].

A virtual screening in ultralarge chemical libraries identified three inhibitors in a first docking study. Then, fragment elaborations gave five promising inhibitors. Finally, hit-to-lead optimization resulted in a compound with an IC_50_ of 0.39 μM for SARS-CoV-2 Mpro (Figure 11) and the crystal structure of the complex led to the design of the most potent one (IC_50_ = 0.077 μM) (Figure 11) [21]. Compounds did not show any effect on cathepsin S activity (IC50 > 50 μM). According to docking, the compound has a hydantoin scaffold, whose carbonyl oxygens form two hydrogen bonds with Asn142 and Cys145.

Two in silico screening studies of compounds from REAL Space or ZINC chemical-compound libraries using two structures of Mpro (PDB id: 6lu7 and PDB id: 6m0k) implemented by a fragment screening study afforded 486 compounds to be synthesized. Then, an in vitro protease-activity assay, performed at 40 μM compound concentration, gave five compounds at a 25% inhibition level [22]. Two of them enhanced the melting temperature of Mpro in a thermal-shift assay, as the known inhibitor GC-376 did. Then, 157 analogs of these two compounds were obtained and tested, affording three compounds that were more active than the parents. Finally, a second round of analog synthesis gave dihydroquinolinone **Z222979552** as the most potent one (Figure 12). The crystal structure of Mpro in complex with compound **Z222979552** showed the compound in the active site not covalently bound to the enzyme (PDB 7P2G). It was also observed that dihydroquinolinone moiety was hydrogen-bonded with Glu166, His163 and His172, and the carbonyl group of the compound hydrogen-bonded the thiol group of Cys145 and the main chain of Glu166. A T-type π-stacking interaction was observed between the benzene ring of the compound and the imidazole ring of His41. Hydrophobic interactions were observed with Asn142, Met49 and Met165 residues. The compound **Z222979552** has antiviral activity in cells and is not toxic.

Indole chloropyridinyl esters were prepared and tested as SARS-CoV-2 Mpro inhibitors by A K. Ghosh et al. Firstly, the compound **GRL-1720** was prepared and tested [23]. It is an irreversible covalent inhibitor of SARS-CoV-2 Mpro with time-dependent inhibition kinetic parameters of k_inact_ = 2.53 ± 0.27 min^−1^, Ki = 2.15 ± 0.49 μM. The IC_50_ value for GRL-1720 after a 10 min incubation is 0.32 μM. A further study to improve indole chloropyridinyl esters analogs as SARS-CoV-2 Mpro inhibitors gave compound **7d** [24]. The mechanism of inhibition of indole chloropyridinyl ester has been studied. The inhibitor covalently modifies SARS-CoV-2 Mpro, forming a thioester bond with the catalytic Cys145 and the indole carbonyl group. The catalytic dyad of Mpro, His41 and Cys145 are involved in the nucleophilic attack on the 5-chloropyridinyl ester to form a tetrahedral intermediate, which then expels the chloropyridinyl group and forms a covalent bond acylating Cys145 in the active site (Figure 13).

In this work, the crystal structure of the complex formed between inhibitor **9d** and SARS-CoV-2 Mpro (PDB 7RC0) showed the sulfur atom of Cys145 to be covalently attached to the indole carbonyl group of **9d** in the S1 pocket. His41 rotates out of the way to π–π stacking with the indole ring (Figure 14) [5].

Recently, the already-reported indole diketopiperazine alkaloids neoechinulin A and echinulin A have been isolated from the Red Sea-derived *Aspergillus fumigatus* MR2012 (Figure 15). Both compounds exhibited inhibitory effects against SARS-CoV-2 Mpro, with IC_50_ values of 0.47 μM and 3.90 μM, respectively [25].

Molecular docking showed the neoechinulin A binding pose, establishing four hydrogen bonds between the diketopiperidine moiety and several Mpro residues from the active site: with Leu141, Asn142 and Gly143 from S1′ site; and with Glu166 from S2 site (Figure 16). Echinulin A established three hydrogen bonds via its diketopiperidine moiety, and the last hydrogen bond with Glu166 was established via its indole NH. Steered molecular dynamics studies indicated that neoechinulin A has the highest binding stability inside the Mpro active site [25].

In a screening study of over 60 natural products, pentagalloyl glucose (PGG) and (-)-epigallocatechin-3-gallate (EGCG) were identified as inhibitors of the main protease of SARS-CoV-2 in the low-micromolar range (Figure 17) [26]. The binding mechanism of these compounds takes place through hydrogen bonds and Van der Waals forces with multiple residues, including those involved in the catalytic activity, as revealed by docking.

A structural, computational and biochemical study identified the natural product shikonin as a SARS-CoV-2 Mpro inhibitor in the low-micromolar range [27]. A crystal structure of the complex formed between SARS-CoV-2 Mpro and shikonin (PDB 7CA8) revealed some interesting differences as compared to previous reported crystal structures (Figure 18) [5]. The catalytic dyad His41-Cys145 underwent dramatic conformational changes. The imidazole ring of residue His41 changed the conformation to accommodate the π–π stacking interaction with the naphthoquinone ring of shikonin. Another large difference was found in a flexible loop of the protease, including Cys44 to Tyr54, Asp187 to Ala191, and Leu141 to Ser144, which were not located in the dimerization region and were irrelevant to crystal packing.

However, as said above, inhibition of Mpro by shikonin was abolished in the presence of DTT, suggesting this compound is a promiscuous inhibitor.

9,10-Dihydrophenanthrene derivatives were tested as SARS-CoV-2 Mpro inhibitors. A Structure–Activity Relationship study resulted in the discovery of the lead compound **C1** (Figure 19). Enzyme kinetic analyses revealed **C1** dose-dependently inhibited Mpro through a mixed-inhibition manner. Molecular-docking simulations elucidated the possible binding mode of **C1** at the dimer interface of the target with the hydroxymethyl group forming a hydrogen-bond interaction with Gln189. Compound **C1** showed good metabolic stability [28].

An investigation of the crystal structure of the noncovalent inhibitor **X77** bound to Mpro of SARS-CoV-2 (PDB code: 6W63) showed that the sulfur atom of Cys145 is positioned at 3.2 Å from the imidazole moiety, suggesting that a covalent warhead could be incorporated. Thus, replacement of the imidazole with a covalent warhead appeared to be a promising strategy to improve the inhibitory potency of this noncovalent inhibitor. The resulting compound could be prepared via a four-component Ugi reaction, enabling a combinatorial approach. Among all prepared compounds, vinyl sulfone **14a** and chloroketone **16a** were the most active, with one order of magnitude more potent than the original **X77** (Figure 20) [29].

Interestingly, metal complexes have also displayed inhibitory activity against SARS-CoV-2 Mpro. Firstly, some Re^1^ tricarbonyl complexes were prepared and tested, with compound **34** being the most active one (Figure 21) [30]. In addition, zinc cation alone has been demonstrated to inhibit the protease in the micromolar range [31]. Then, zinc thiotropilone complexes were reported to be Mpro inhibitors, giving IC_50_ with nanomolar values [32] (Figure 21). Bismuth drugs, colloidal bismuth subcitrate (CBS), alone or in combination with N-acetyl-L-cysteine (NAC), inhibited SARS-CoV-2 Mpro with an IC_50_ of 21.10 μM and 22.25 μM, respectively (Figure 21) [33].

In summary, new molecules acting as inhibitors of the main protease (Mpro) of SARS-CoV-2 virus have been reported. Some of them show a short peptide structure, usually with a glutamine surrogate at P1 site, a leucine-like residue at the P2 site, and an electrophilic group at the carboxyl end to react with cysteine 145. Other Mpro inhibitors are nonpeptidic compounds, namely cyclic compounds or natural products. Metal complexes have been also reported as Mpro inhibitors. The compounds shown in this review show low/submicromolar activity in both in vitro and in vivo models and could lead to promising drug candidates for COVID-19 disease. So far, one inhibitor has been commercialized and is currently used against COVID-19 disease in combination with another drug. It is expected that more Mpro inhibitors rationally designed based upon accumulated structural information will come out and become better drugs in the near future. This huge effort carried out by the scientific community will also benefit the discovery of drugs for other diseases.

## Figures and Tables

**Figure 1 molecules-27-02523-f001:**
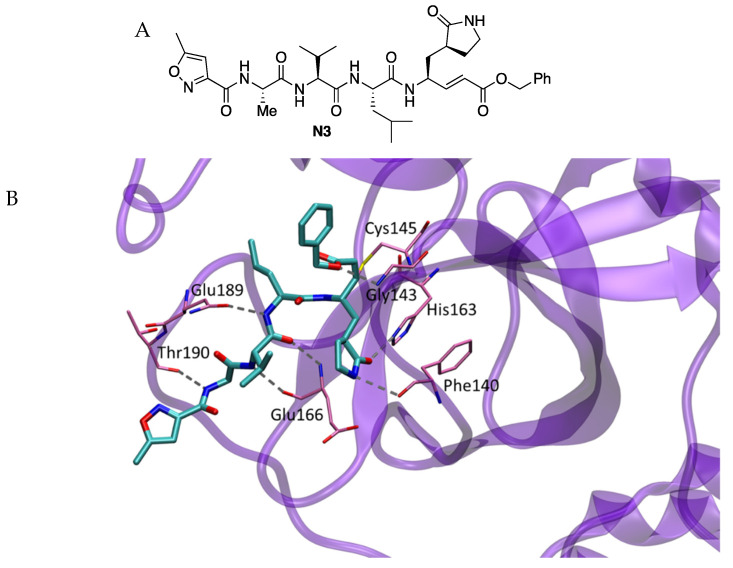
(**A**) Chemical structure of **N3** inhibitor. (**B**) X-ray structure of the complex between **N3** and SARS-CoV-2 Mpro.

**Figure 2 molecules-27-02523-f002:**
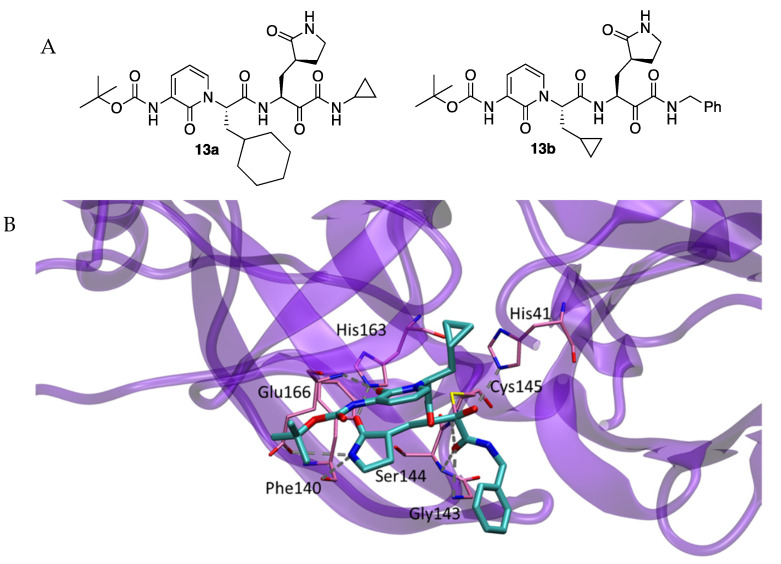
(**A**) Inhibitors **13a** and **13b**. (**B**) Details of X-ray structure of SARS-CoV-2 Mpro in complex with inhibitor **13b**.

**Figure 3 molecules-27-02523-f003:**
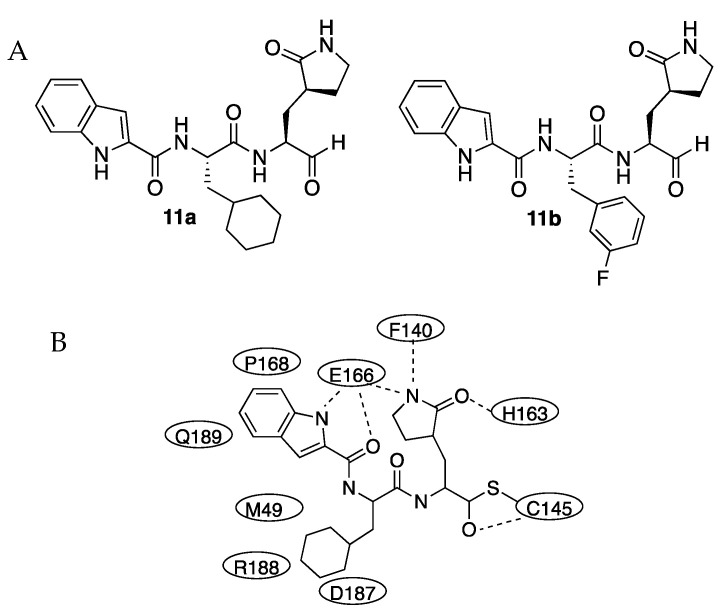
(**A**) Inhibitors **11a** and **11b**. (**B**) Scheme of the interactions in the X-ray structure of inhibitor **11a** with SARS-CoV-2 Mpro.

**Figure 4 molecules-27-02523-f004:**
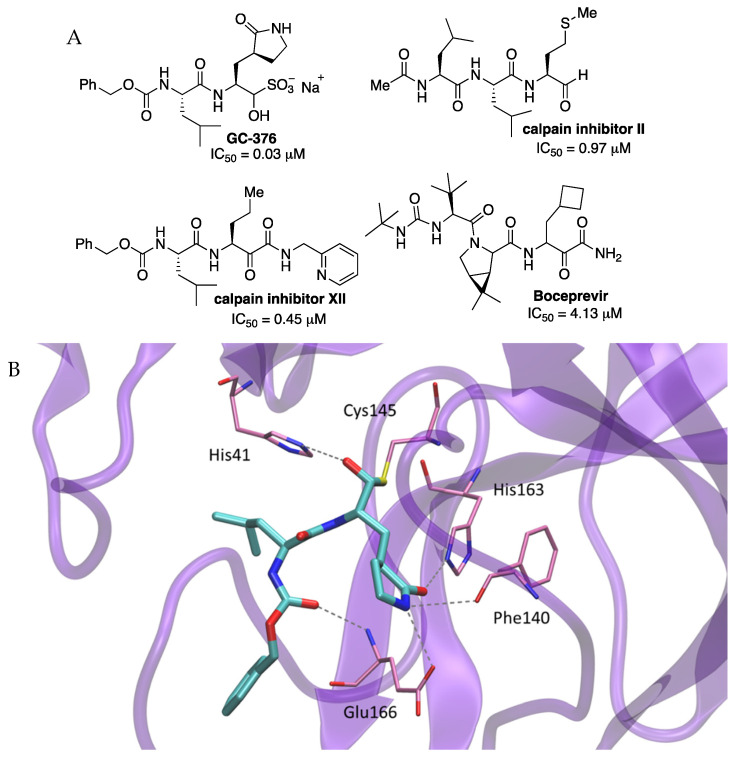
(**A**) Inhibitors of SARS-CoV-2 Mpro enzyme. (**B**) X-ray structure of the complex between GC-376 and SARS-CoV-2 Mpro.

**Figure 5 molecules-27-02523-f005:**
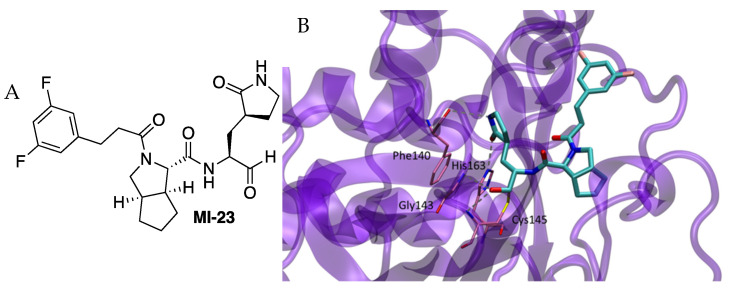
(**A**) Chemical structure of Inhibitor **MI-23**. (**B**) Detail of x-ray structure.

**Figure 6 molecules-27-02523-f006:**
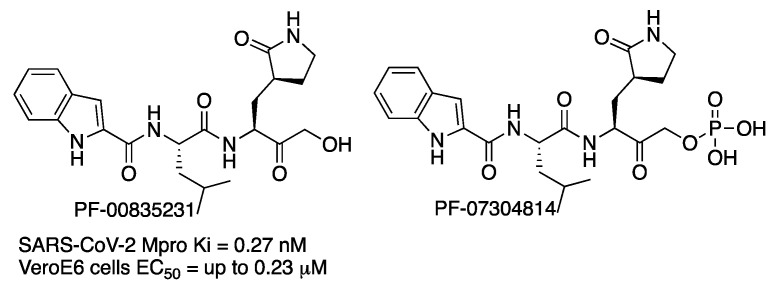
Pfizer inhibitors of SARS-CoV-2 Mpro enzyme.

**Figure 7 molecules-27-02523-f007:**
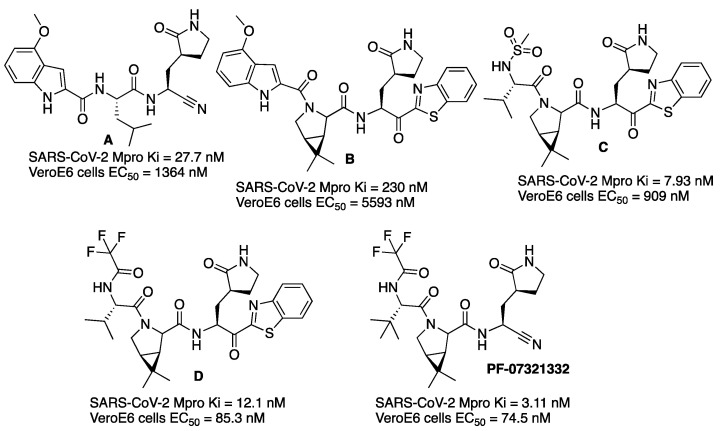
Pfizer inhibitors of SARS-CoV-2 Mpro enzyme.

**Figure 8 molecules-27-02523-f008:**
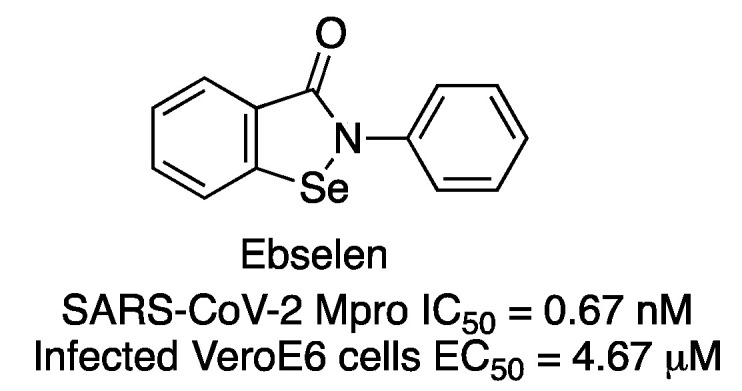
Chemical structure and inhibitory data of ebselen.

**Figure 9 molecules-27-02523-f009:**
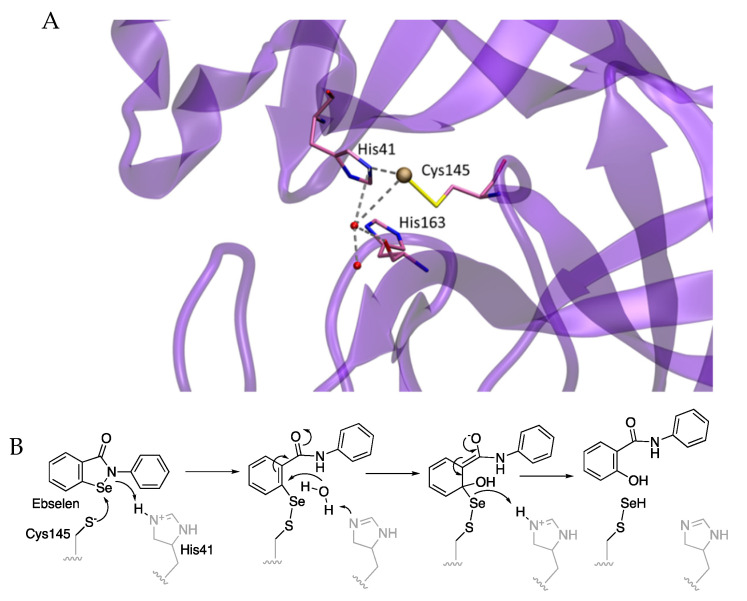
(**A**) Detail of crystal structure of selenium–Mpro complex. (**B**) Mechanism of ebselen.

**Figure 10 molecules-27-02523-f010:**
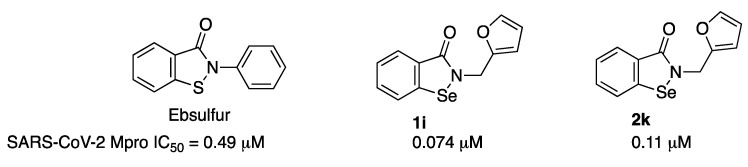
Ebsulfur and ebselen derivatives.

**Figure 11 molecules-27-02523-f011:**
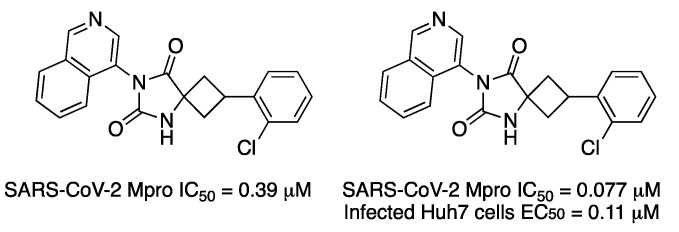
SARS-CoV-2 Mpro inhibitors with a hydantoin scaffold.

**Figure 12 molecules-27-02523-f012:**
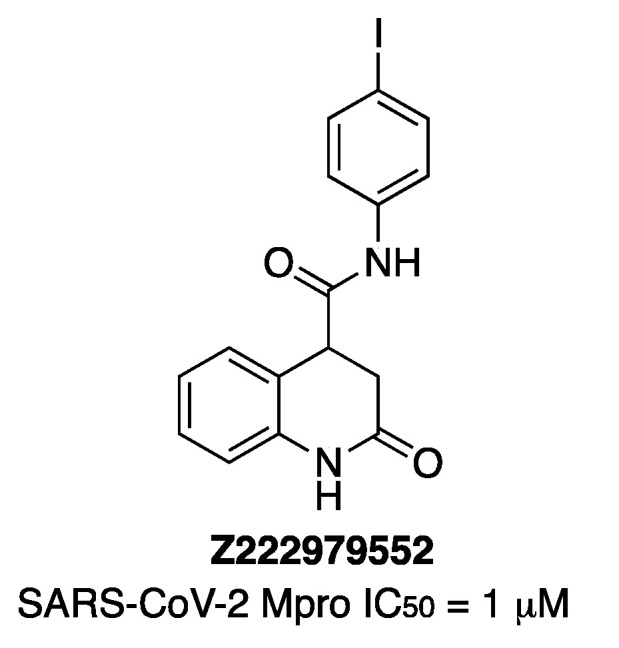
Chemical structure of compound **Z222979552**.

**Figure 13 molecules-27-02523-f013:**
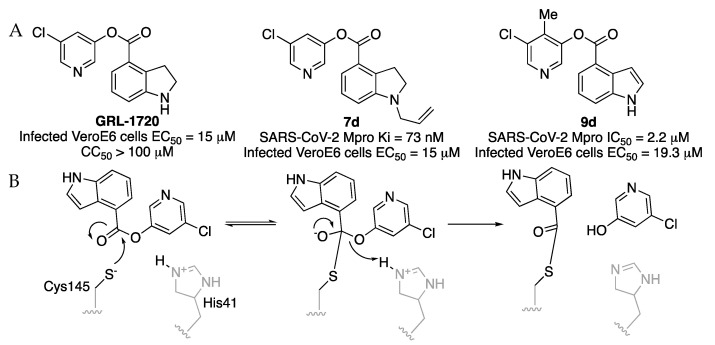
(**A**) Indole chloropyridinyl esters as SARS-CoV-2 Mpro inhibitors. (**B**) Mechanism of inhibition.

**Figure 14 molecules-27-02523-f014:**
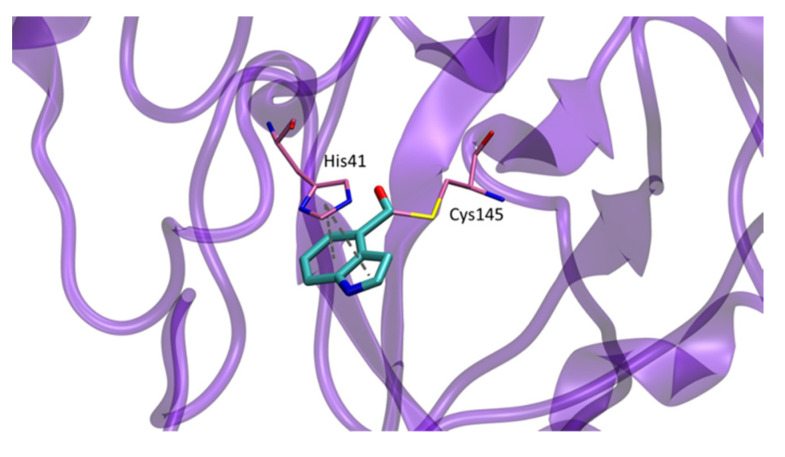
Detail of crystal structure of Mpro-**9d** complex.

**Figure 15 molecules-27-02523-f015:**
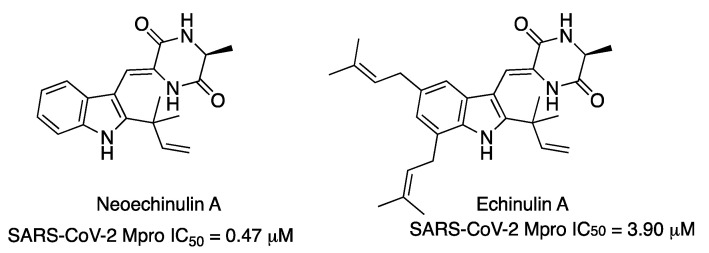
Marine natural products neoechinulin A and echinulin A inhibitors of SARS-CoV-2 Mpro enzyme.

**Figure 16 molecules-27-02523-f016:**
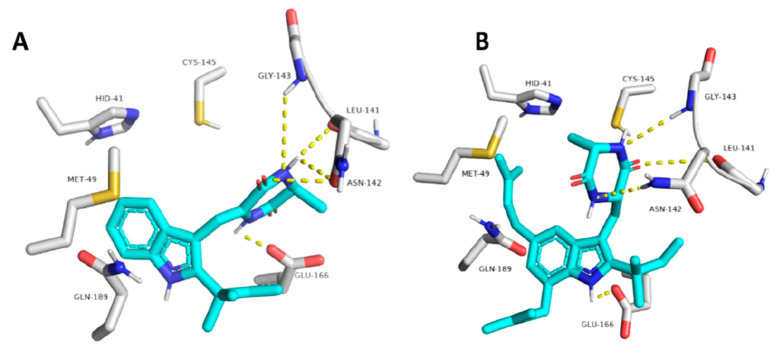
Binding poses of neoechinulin A and echinulin A into the active site of SARS-CoV-2 Mpro enzyme (**A** and **B**, respectively, images from [25]).

**Figure 17 molecules-27-02523-f017:**
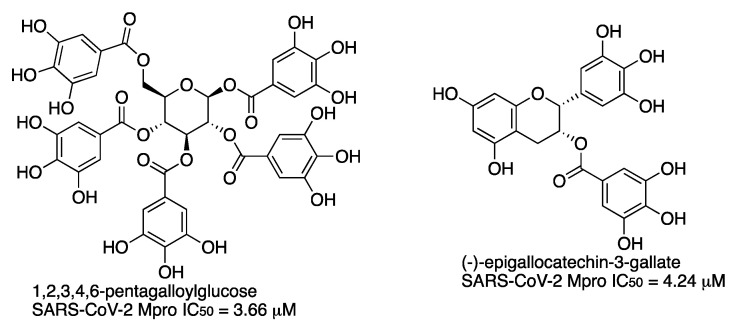
Natural products pentagalloyl glucose (PGG) and (-)-epigallocatechin-3-gallate (EGCG) as SARS-CoV-2 Mpro inhibitors.

**Figure 18 molecules-27-02523-f018:**
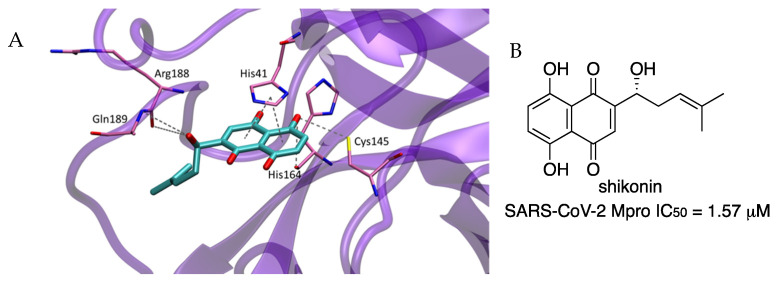
(**A**) Detail of crystal structure of the complex formed between shikonin and SARS-CoV-2 Mpro. (**B**) Chemical structure of shikonin.

**Figure 19 molecules-27-02523-f019:**
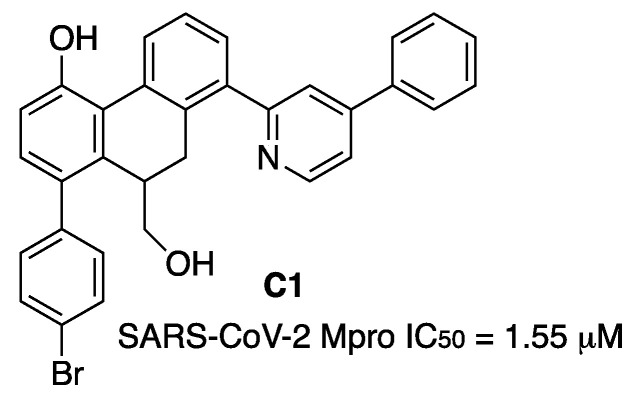
Dihydrophenanthrene **C1**.

**Figure 20 molecules-27-02523-f020:**
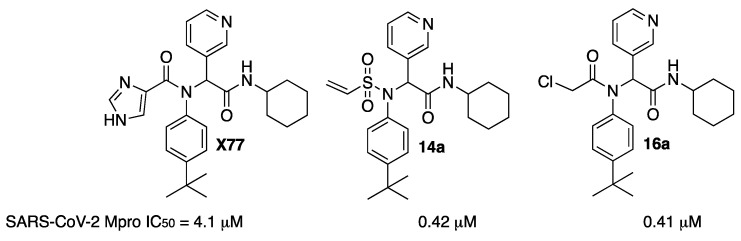
Inhibitors **14a** and **16a** derived from **X77**.

**Figure 21 molecules-27-02523-f021:**
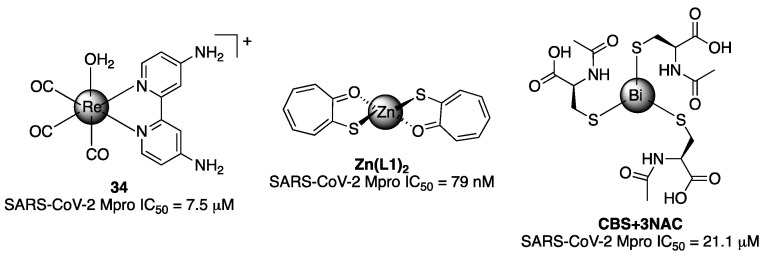
Metal complexes as inhibitors of SARS-CoV-2 main protease.

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
