# Peer review of "Advances in the Development of SARS-CoV-2 Mpro Inhibitors"

_molecules, 2022, doi:10.3390/molecules27082523_

Round 1

Reviewer 1 Report

The authors present an exhaustive review the recent development on inhibitors of the main protease (Mpro) of virus SARS-CoV-2 focusing on peptide and chemical inhibitors, which is comprehensive, well structured review and of interest for the targeting Mpro to control viral infection.  

Minor comments:

  1. Figures: The separate presentations of the individual structure and target of the inhibitor could be merged in one figure.
  2. Perspective: The review would benefit of a conclusion and perspective of the novel inhibitors in controlling SARS-CoV-2 virus infection and future development.

Author Response

Please find below point-by-point responses to the reviewer's comments:

Reviewer: "The authors present an exhaustive review the recent development on inhibitors of the main protease (Mpro) of virus SARS-CoV-2 focusing on peptide and chemical inhibitors, which is comprehensive, well structured review and of interest for the targeting Mpro to control viral infection.  

Minor comments:

  1. Figures: The separate presentations of the individual structure and target of the inhibitor could be merged in one figure.
  2. Perspective: The review would benefit of a conclusion and perspective of the novel inhibitors in controlling SARS-CoV-2 virus infection and future development."

Response: We appreciate the suggestions made by the reviewer. All the changes have been made as suggested.

Author Response

Please find below the point-by-point responses to the reviewer:

Reviewer: " This manuscript comprehensively summarized recent progress on the development of inhibitors of Mpro form SARS-CoV-2 and their potential mechanism of actions based on X-ray crystallography studies. In terms of inhibitors, the author mainly focused on a peptide display an adequate sequence for high affinity and a reactive warhead. The second category is a diverse group of organic molecules that do not have a peptide framework. However, the authors neglect an important group of metal based inhibitors such as zinc, bismuth (which was demonstrated potential drug for the treatment of COVID-19). I strongly suggest the author should include these works as listed below: 

Yuan, S. et al. Metallodrug ranitidine bismuth citrate suppresses SARS-CoV-2 replication and relieves virus-associated pneumonia in Syrian hamsters. Nat Microbiol 5, 1439-1448, doi:10.1038/s41564-020-00802-x (2020). 

Wang, R. M. et al. Orally administrated bismuth drug together with N-acetyl cysteine as a broad-spectrum anti-coronavirus cocktail therapy. Chem Sci, doi:10.1039/D1SC04515F (2021). 

Panchariya, L. et al. Zinc(2+) ion inhibits SARS-CoV-2 main protease and viral replication in vitro. Chem Commun 57, 10083-10086, doi:10.1039/d1cc03563k (2021). 

DeLaney, C. et al. Zinc thiotropolone combinations as inhibitors of the SARS-CoV-2 main protease. Dalton T 50, 12226-12233, doi:10.1039/d1dt02499j (2021). 

Karges, J., Kalaj, M., Gembicky, M. & Cohen, S. M. Re-I Tricarbonyl Complexes as Coordinate Covalent Inhibitors for the SARS-CoV-2 Main Cysteine Protease. Angew Chem Int Edit 60, 10716-10723, doi:10.1002/anie.202016768 (2021). "

Response: We appreciate the suggestion made by the reviewer. A new paragraph and a new figure have been added in the manuscript to explain complexes of Re, Zn and Bi as inhibitors of SARS-CoV-2 Mpro. All suggested references have been added except for the first one because it is not about Mpro inhibitors.

Reviewer: " In general, the review is well written and deserved to be published after amendment the point raised. 

Some minor correction is necessary to improve the quality of the review as detailed below: 

P39 the coronavirus express…. the coronaviruses 

P42 the main protease (Mpro) also known….. is also known 

P67 … was identified… was resolved 

P95-P100 to be consistent with others in the manuscript, please change the subscript e.g. Cys145 to Cys145 etc. 

P109, Detail of x-rays complex of inhibitor 13b with SARS-CoV-2 Mpro -change to Details of X-ray structure of SARS-CoV-2 Mpro in complex with inhibitor 13b 

P138: the x-rays complex… change to X-ray structure 

Figure 8 and 9 should be redrawn to make the labels larger to be consistent with other figures 

P199 x-rays structure … change to X-ray structure 

Figure 20 should be redrawn to improve the quality and enlarge the labels to make them readable"

Response: We appreciate the suggestion made by the reviewer. All the changes have been made as suggested.